# Influence of Thermal and Flash-Lamp Annealing on the Thermoelectrical Properties of Cu_2_ZnSnS_4_ Nanocrystals Obtained by “Green” Colloidal Synthesis

**DOI:** 10.3390/nano13111775

**Published:** 2023-05-31

**Authors:** Yevhenii Havryliuk, Volodymyr Dzhagan, Anatolii Karnaukhov, Oleksandr Selyshchev, Julia Hann, Dietrich R. T. Zahn

**Affiliations:** 1Semiconductor Physics, Chemnitz University of Technology, 09107 Chemnitz, Germany; 2Center for Materials, Architectures and Integration of Nanomembranes (MAIN), 09126 Chemnitz, Germany; 3V.E. Lashkaryov Institute of Semiconductor Physics NAS of Ukraine, 03028 Kyiv, Ukraine; 4Physics Department, Taras Shevchenko National University of Kyiv, 01601 Kyiv, Ukraine

**Keywords:** CZTS NCs, Cu_2_ZnSnS_4_, nanocrystals, flash-lamp annealing, Raman spectroscopy, thermoelectric properties, Seebeck measurements, conductivity, secondary phases, Cu_x_S

## Abstract

The problem with waste heat in solar panels has stimulated research on materials suitable for hybrid solar cells, which combine photovoltaic and thermoelectric properties. One such potential material is Cu_2_ZnSnS_4_ (CZTS). Here, we investigated thin films formed from CZTS nanocrystals obtained by “green” colloidal synthesis. The films were subjected to thermal annealing at temperatures up to 350 °C or flash-lamp annealing (FLA) at light-pulse power densities up to 12 J/cm^2^. The range of 250–300 °C was found to be optimal for obtaining conductive nanocrystalline films, for which the thermoelectric parameters could also be determined reliably. From phonon Raman spectra, we conclude that in this temperature range, a structural transition occurs in CZTS, accompanied by the formation of the minor Cu_x_S phase. The latter is assumed to be a determinant for both the electrical and thermoelectrical properties of CZTS films obtained in this way. For the FLA-treated samples, the film conductivity achieved was too low to measure the thermoelectric parameters reliably, although the partial improvement of the CZTS crystallinity is observed in the Raman spectra. However, the absence of the Cu_x_S phase supports the assumption of its importance with respect to the thermoelectric properties of such CZTS thin films.

## 1. Introduction

The development of alternative energy sources has been increasing in extent in recent decades. The importance of clean and renewable energy sources is increasing not only because of the need for environmental protection but also because of the growing demand for various portable energy sources. One of the most attractive renewable energy sources is solar energy. The extensive development of photovoltaics has enabled not only a move from “traditional” Si-based solar panels to other materials with conversion efficiencies above 20% [1,2] but also the use of flexible [3] and transparent [4] solar panels in everyday life. However, one of the main issues with solar panels is their overheating, as most of the absorbed solar energy is lost as waste heat. This reduces the efficiency of the solar panels and can even cause damage to them [5]. There are various ways of cooling solar panels or/and using this heat [6,7,8], but the most promising way is to create hybrid photovoltaic–thermoelectric devices, in which this heat is directly converted into electricity [9,10,11,12,13]. This explains the increasing attention of researchers to materials possessing thermoelectric and photovoltaic properties simultaneously.

The most efficient materials for thermoelectric conversion are BiTe, SnSe, and CuSe [14]. However, these materials are expensive or/and environmentally unfriendly and do not have pronounced photovoltaic properties. Therefore, there are significant efforts dedicated to searching for materials that possess photovoltaic and thermoelectric properties and are cheap, easy to produce, and environmentally friendly. One such material system is the Cu_2_ZnSnS_4_ (CZTS) family. These materials are well known for their photovoltaic applications [15,16,17,18] and the ability to produce CZTS nanocrystals (NCs) by various methods [19], especially “green” synthesis in colloidal solutions [15,16,20]. This makes these materials even more attractive and promising for use in so-called third-generation photovoltaics. Recently, CZTS has also attracted attention in terms of its thermoelectric properties [21,22,23,24,25,26,27,28,29,30]. Both bulk crystals [26], polycrystalline layers [25,27], and nanocrystals [22] are considered. Controllable deviations from the Cu_2_ZnSnS_4_ stoichiometry [23] and cation substitution [28] were suggested as parameters to be tuned for the optimization of the CZTS TE performance [27]. Another interesting finding of those works is that the cation disorder and the presence of secondary phases in CZTS-like compounds, which deteriorate the photovoltaic conversion, may turn out to be advantageous for thermoelectric properties [24,31,32,33]. This fact may enable an acceptable overall efficiency of the hybrid device. Therefore, a deeper insight into the dependence of thermoelectric properties on CZTS NC film fabrication is needed.

One of the most common methods to tune the properties of semiconductor films is thermal annealing. For CZTS, this is known as an efficient tool not only to improve the crystallinity but also to induce transitions between disordered and ordered kesterite structures, which was shown to occur at 260 °C [34]. The resulting phase plays an important role not only in the annealing temperature but also in the cooling rate. In particular, rapid cooling (quenching) fixes the cation-disordered phase obtained at high temperatures [31]. However, due to the presence of copper in the compound, which may easily oxidize, the thermal annealing of CZTS NCs requires an inert or vacuum atmosphere. However, even under these conditions, the formation of a secondary Cu_x_S phase can take place, especially at annealing temperatures above 200 °C [35].

An alternative method of thermal material treatment is flash-lamp annealing (FLA). It has been known for decades but started to gain popularity with the growing application of thin-film technologies [36,37,38,39]. FLA was efficiently used to improve the homogeneity and crystallinity of thin films [40] or to make multicomponent compounds from binary ones [41]. The advantage of this method is not only its scalability and speed but also that it does not require a special atmosphere due to the very short annealing time (milliseconds). We previously studied the effect of FLA in a wide range of energy densities up to 60 J/cm^2^ on the structure of CZTS NCs [41] and observed secondary phase formation and crystallinity changes in CZTS NCs as a result of FLA.

The thermoelectric efficiency of a material can be described by the figure of merit $ZT=\frac{\sigma S^{2}}{k}T$, where *ZT* is a dimensionless quantity, *S* is the Seebeck coefficient, *σ* is the electrical conductivity, *k* is the thermal conductivity, and *T* is the temperature. Thus, one would expect that the transition from bulk materials to NCs would result in an improvement in *ZT* due to a reduction in thermal conductivity caused by additional heat dissipation at the NC boundaries. The second quantity used to characterize thermoelectric materials is the so-called power factor (*PF*), $PF=\sigma S^{2}$. The information provided about the thermoelectric properties of the material is not complete but is sufficient to obtain an idea of the thermoelectric efficiency.

Here, we investigated the effect of thermal annealing and FLA treatment on CZTS NCs obtained by green synthesis in colloidal solutions using a small ligand, namely, thioglycolic acid (TGA). Based on our previous studies on the effect of FLA treatment of CZTS-TGA NCs [41], FLA energy densities less than 12 J/cm^2^ were chosen for this work, since the formation of ternary secondary phases and even the deterioration of CZTS NCs to binary and ternary compounds were observed when using higher values. The effect of the two types of annealing on the film morphology was investigated using atomic force microscopy (AFM). Structural changes were studied using Raman spectroscopy, which has proved to be a very informative and useful tool for the characterization of CZTS [42,43,44,45]. While the dependence of the thermoelectric properties of CZTS NCs on thermal annealing was partially studied in the literature for NCs obtained by other methods of synthesis [22,31,46] and the effect of FLA on the CZTS-TGA NCs composite with PEDOT:PSS was recently studied by our group [47], the dependence of the thermoelectric properties of CZTS-TGA NCs on thermal and FLA treatments has not yet been reported.

## 2. Materials and Methods

The synthesis of CZTS NCs was carried out following the procedure that we previously described [48,49]. After the synthesis, the NC solution was purified from residual reaction products and excess ligands by adding isopropanol and centrifuging the solution. Afterward, the precipitated CZTS NCs were redissolved in deionized water. Spin coating was performed using a Laurell Spin Coater WS-650MZ-23NPP/LITE (Laurell Technologies Corporation, Lansdale, PA, USA). The solution was spin-coated for 20 s with a rotation speed of 2000 rpm and acceleration of 500 rpm/s onto glass substrates cleaned with ethanol and deionized water.

The surface characterization was performed using an Agilent 5500 AFM (Agilent Technologies, Inc., Santa Clara, CA, USA) in the acoustic AFM (ACAFM) mode.

Flash-lamp annealing was carried out using an FLA-D100PA flash-lamp annealing system from Dresden Thin Film Technology GmbH (SOLAYER GmbH, Kesselsdorf, Germany) based on a Xenon lamp emitting in the range of 300 to 800 nm in a glove box system with a nitrogen atmosphere. The values of the power density that we used (2 J/cm^2^, 6 J/cm^2^, etc.) are known from its specifications. As the illuminated area (≈40 × 750 mm) is much larger than the size of our samples (≈4 × 25 mm), the illumination is expected to be homogeneous for each sample.

Thermal annealing was performed in a nitrogen atmosphere in a glove box with a heating plate (Heidolph MR 3001 K; Heidolph Instruments GmbH & Co. KG, Schwabach, Germany).

Raman spectra were excited using a 514.7 nm solid-state laser (Cobolt; HÜBNER GmbH & Co. KG, Kassel, Germany) and were registered at a spectral resolution of about 1 cm^−1^ using a LabRam HR800 equipped with cooled CCD detectors (HORIBA Jobin Yvon GmbH, Bensheim, Germany). The incident laser power under the microscope objective (100×) was in the range of 0.01–0.1 mW.

Thermoelectric and electric characterizations were performed using an “IPM-SR7 Power factor measurement system” customized for films and bulk materials in the temperature range of 300 to 900 K (Fraunhofer IPM, Freiburg, Germany). For the thermoelectric measurements, the samples were prepared using spin-coating deposition onto glass substrates. Glass was chosen as a substrate due to its low thermal and electrical conductivities, minimizing the influence of the substrate on measurements. The characterization was performed at 310 K to avoid any influence of the measurement conditions on the structure and clearly separate the influence of different annealing conditions. To correctly determine the electrical conductivity, the thickness of each sample was measured using AFM. In this process, the AFM profile was measured on an edge, and the height was considered to be the thickness of the film. The films had thicknesses measured in the range of 100–200 nm. The range was influenced mainly by the removal of additional moisture from the films during annealing.

Aqueous solutions of CZTS NCs obtained by green colloidal synthesis were deposited on glass substrates by spin coating. After that, the samples were divided into two groups, one for thermal annealing and the other for FLA treatment. Thermal annealing was carried out in the temperature range from 50 °C to 350 °C with a step of 50 °C and an annealing time of 15 min. In order to prevent the oxidation of the samples, annealing was carried out in a nitrogen atmosphere in a glove box. After annealing, the samples were naturally cooled down and removed from the glove box for analysis.

The FLA treatment was carried out in the range of energy densities from 2 J/cm^2^ to 12 J/cm^2^ with a step of 2 J/cm^2^. The FLA duration was 17.8 ms. The FLA treatment was carried out in a nitrogen atmosphere in a glove box.

## 3. Results and Discussion

### 3.1. Surface Morphology

After the annealing procedure, surface characterization was carried out using AFM and Gwyddion 2.61 software to determine the surface roughness. For this purpose, a polynomial background was extracted from the resulting AFM images, and the Gwyddeon “statistical quantities” tool was applied. Figure 1 shows the AFM images for the thermally annealed samples. With increasing annealing temperature, a slight decrease in surface roughness is observed, from 52.9 ± 0.1 nm for the unannealed sample to 45.2 ± 0.1 nm for the one annealed at 50 °C and down to 36.9 ± 0.1 nm for the film annealed at 200 °C. This can be attributed to the removal of residual moisture and the partial removal of ligands, resulting in a denser arrangement of NCs. It is unlikely that temperatures in this range are sufficient to sinter individual NCs into microcrystals. At annealing temperatures of 250 °C and above, an increase in surface roughness is observed. This coincides with the appearance of features in Raman spectra that correspond to the secondary Cu_x_S phase, as shown in Section 3.2. Therefore, the increase in surface roughness at 250 °C may be directly related to the formation of this secondary phase, which tends to form on the CZTS surface [50,51,52,53]. It is important to note that for the sample annealed at 350 °C, the presence of the Cu_x_S phase is not observed in Raman spectra; nevertheless, the roughness is much higher (78.8 nm) than for all other samples. This phenomenon may be explained by the fact that a temperature of 350 °C is sufficient for CZTS crystallites to start merging into larger crystallites, and this process is more favorable than Cu_x_S phase segregation. This is confirmed not only by the increased roughness but also by the surface heterogeneity, which is noticeable in the AFM images.

The AFM images of FLA-treated samples are shown in Figure 2. At low annealing energy densities, a slight decrease in surface roughness to 49.1 ± 0.1 nm at 6 J/cm^2^ is observed. This fact is consistent with literature data, where FLA with low energy densities was used to sinter individual crystallites to obtain a more homogeneous film [40]. Starting from 8 J/cm^2^ and higher, it is observed that the material starts to form “islands”, on top of which the roughness value decreases even more, but the overall roughness increases to around 54–56 nm, and the film becomes less homogeneous due to the formation of such islands. This behavior can be explained by the fact that at an energy density of FLA above 8 J/cm^2^, small crystallites start to sinter and agglomerate into larger crystallites.

### 3.2. Raman Study

All samples obtained were further analyzed by Raman spectroscopy. The spectra of thermally annealed and FLA-treated samples are shown in Figure 3a,b, respectively. The position of the Raman characteristic band for the untreated sample is 332 ± 1 cm^−1^, with an FWHM of 36 ± 1 cm^−1^, which is typical for CZTS NCs obtained by colloidal synthesis and corresponds to the structure of disordered kesterite, in accordance with our earlier results on similarly synthesized NCs [54]. The line shape with broad features located at 200–300 cm^−1^ and around 370 cm^−1^ is also consistent with literature data for the disordered kesterite structure of bulk CZTS [55] and CZTS NCs [56,57]. Upon the thermal annealing of CZTS NCs up to 200 °C, a gradual reduction in the FWHM down to 27 ± 1 cm^−1^ is observed. Since the position of the peak remains unchanged, this decrease in the half-width can be interpreted as a decrease in the number of defects without a change in the type of structure. At an annealing temperature of 250 °C, a high-frequency shift of the CZTS characteristic Raman band to 337 ± 1 cm^−1^ and a significant decrease in the FWHM to 18 ± 1 cm^−1^ are observed. On the one hand, such changes may be evidence for a transition to an ordered kesterite structure, because it is known to have its Raman peak position at 337–339 cm^−1^ [42,45,58,59,60]. However, other possibilities, such as the formation of a secondary phase and an accompanying change in CZTS stoichiometry, cannot be completely excluded. The spectral feature around 473 ± 1 cm^−1^ indicates the formation of Cu_x_S [49,54], while the large FWHM of this band indicates the dispersion of its composition *x* and low structural perfection or crystallite size of this secondary phase. As the formation of Cu_x_S consumes some of the copper ions, it may stimulate the formation of secondary phases consisting of other cations, i.e., ZnS, Sn-S, and Zn-Sn-S. We did not observe any Raman features for the former two binary phases using green excitation, expected at 150–250 cm^−1^ for SnS and 240 cm^−1^ for ZnS [49]. Zn-Sn-S NCs synthesized under similar synthesis conditions were found to exhibit the main characteristic band around 340 cm^−1^ [49]. The overlap of this band with the 332 ± 1 cm^−1^ band of CZTS NCs could result in a band at 337 ± 1 cm^−1^ for the sample annealed at 250 °C. However, the band of the secondary Zn-Sn-S phase should have not only a higher frequency but also a large FWHM, and the width of the resulting overlap with the CZTS peak would give a broader Raman band, while we observed peak narrowing for this sample (Figure 3a). Therefore, the upward shift of the CZTS mode after annealing at 250 °C most likely indicates the conversion of CZTS into ordered kesterite with improved stoichiometry, accompanied by the formation of a minor amount of the secondary Cu_x_S phase (probably from the excess Cu left from the structural conversion of CZTS). In the spectrum of the sample annealed at 300 °C, we can observe the presence of the CZTS NC band with the same frequency and half-width as for 250 °C annealing, but with almost zero intensity of the Cu_x_S band (Figure 3a).

It is noteworthy that with a further increase in the annealing temperature, the Raman band position and width return to those obtained for annealing temperatures below 250 °C. Therefore, one can make a preliminary assumption that for this sort of NC, thermal annealing at temperatures around 200–250 °C is optimal for the stabilization of the ordered kesterite phase, while the concomitant formation of the Cu_x_S phase may be the reason for the high electrical conductivity of the films, which also allows the TE parameters of the films to be determined, as discussed below (Figure 4).

Figure 3b shows Raman spectra for FLA-treated samples at different energy densities. One can observe a slight gradual shift of the main CZTS band from 332 ± 1 cm^−1^ to 333 ± 1 cm^−1^, as well as a reduction in the half-width from 36 ± 1 cm^−1^ to 24 ± 1 cm^−1^ with increasing FLA energy density and no trace of secondary phases. This indicates an improvement in the crystallinity of CZTS NCs and is in full agreement with our previous studies on the effect of FLA on CZTS NCs [41].

### 3.3. Thermoelectrical Characterization

Both series of samples were then investigated for their thermoelectric properties, namely, electrical conductivity, the Seebeck coefficient, and the power factor. Figure 4 shows the experimental results of thermoelectric measurements as a function of annealing temperature. At annealing temperatures ranging up to 200 °C, the electrical conductivity was below the detection limit of the device. A slight upward shift of the Raman peak position in this annealing range corresponds to the improvement of crystalline quality, as discussed above. However, this improvement of the crystallinity does not lead to a noticeable change in the conductivity. Such a low conductivity of CZTS NC films annealed at low temperatures was already observed for CZTS NCs synthesized and purified in a way similar to this work [35]. For annealing temperatures of 250 °C and 300 °C, a steep increase in electrical conductivity is observed, which could be due to the formation of a minor quantity of the Cu_x_S phase [35], accompanying the transformation of disordered to ordered kesterite, as discussed in the Raman part above. As drastic changes in the structure and electrical conductivity of the films annealed at 250 °C and 300 °C occur, the conductivity of such films is sufficient for proper thermoelectrical measurement. So, both the Seebeck coefficient (*S*) and power factor (*PF*) exhibit increasing trends, from *S* = 16 ± 0.75 µV/K and *PF* = 9.93 × 10^−3^ ± 1.49 × 10^−3^ µW/cmK^2^ for the sample annealed at 250 °C to *S* = 30 ± 1.5 µV/K and *PF* = 4.724 × 10^−2^ ± 7.09 × 10^−3^ µW/cmK^2^ for the one annealed at 300 °C. The reason for the increase in the electrical conductivity of the sample annealed at 300 °C with the simultaneous disappearance of the pronounced Cu_x_S band in Raman spectra could be a change in the composition of the Cu_x_S phase, where 1 ≤ x ≤ 2 [35,61]. In particular, a more S-poor composition may significantly reduce the intensity of the Raman peak that is due to S-S vibrations [61]. There is probably only a slight variation in the composition x, which can be around 1.8, which would have almost no influence on the electrical conductivity [62]. At the same time, the increase in the Seebeck coefficient may be related to a transition of the CZTS NC phase from more ordered to disordered kesterite, which is more preferable for thermoelectric properties [31,32,33]. This is supported by the slight downward shift of the CZTS characteristic Raman band from 337 to 336 cm^−1^ (Figure 4). So, a clear correlation between the presence of the Cu_x_S phase, the position of the CZTS Raman peak, and the thermoelectrical properties of the CZTS NC thin films is observed. The transition to a more ordered CZTS NC structure, together with the appearance of the secondary Cu_x_S phase, leads to the significant improvement of electrical properties, while an increasing amount of disorder in CZTS NCs in the presence of the Cu_x_S phase leads to the improvement of thermoelectrical properties. The thermoelectrical values obtained are in good agreement with literature data for pure [62] and doped [63] Cu_x_S. Therefore, we can assume that the secondary Cu_x_S phase formed on the surface of the CZTS NC film [50,51,52,53] is responsible not only for the conductivity but also for the thermoelectrical properties of such a film. The reasons why this secondary phase appears only in a limited range of annealing temperatures require further detailed studies.

The obtained electrical conductivity values for the FLA-treated samples up to 10 J/cm^2^ were in the range of the error bar, and the conductivity of such films was not sufficient for proper thermoelectrical measurements. The slight increase in conductivity at low energy densities may be due to an increase in film homogeneity (Figure 2a,b) due to sintering between individual crystallites. The increase in conductivity for an energy density of 12 J/cm^2^ may be understood, based on our results for the thermally annealed samples, as the beginning of the formation of a secondary Cu_x_S phase, the amount of which is not yet sufficient to be detected in the Raman spectra or make any significant changes to the TE properties of the film. However, even at this value of electrical conductivity, it was not possible to reliably obtain the Seebeck coefficient.

In general, for all films of CZTS NCs obtained by the colloidal synthesis in this work, the conductivity is much lower than the values commonly reported for bulk crystals [64] or polycrystalline films obtained by other methods [31]. The main reason may be the poorer crystallinity of our films due to the mild conditions of both the NC synthesis and film fabrication. In addition, the films may contain some minor residues of ligands not completely removed by the centrifugation of the colloid or the annealing of the film. The component composition of CZTS NCs may also play an important role, as well as the thickness and quality of the deposited films. Thus, to improve the conductivity of the CZTS NC films obtained by such a synthesis method, the synthesis conditions should be further optimized (for example, further purification from ligands, different NC composition, etc.), as well as the film deposition method (thicker and denser films are expected to show better conductivity).

## 4. Conclusions

Thin films (~150 nm) of colloidal CZTS NCs (diameter ~10 nm) synthesized by a facile (“green”) colloidal route in water were subjected to thermal annealing at various temperatures or FLA treatment at different energy densities. A correlation between surface roughness changes and the improvement of the crystal quality of CZTS NC thin films treated by FLA and thermal annealing was observed. The very low conductivity of CZTS NC films obtained by spin coating from the “green” colloidal solution makes the determination of the thermoelectrical parameters very challenging. At annealing temperatures of 250 °C and 300 °C, a secondary Cu_x_S phase is detected in the Raman spectra. This phase, which is most probably detrimental to the photovoltaic effect, is found to be responsible for the improvement of the electrical and thermoelectric performance of the NC films. For the FLA-treated samples, no sufficient improvement of the conductivity could be achieved so that TE parameters could be measured, although the partial improvement of the CZTS crystallinity is observed in the Raman spectra. We can assume that due to the short duration (ms) of FLA pulses, the CZTS lattice may not have enough time to undergo the structural transition and separation of Cu_x_S.

## Figures and Tables

**Figure 1 nanomaterials-13-01775-f001:**
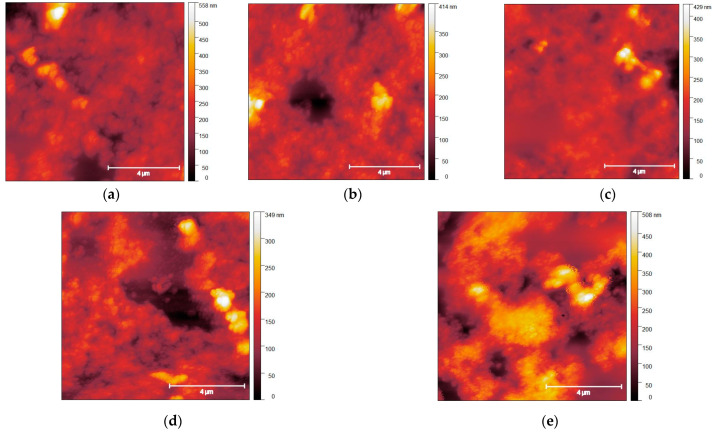
Typical AFM topography images of (**a**) unannealed film and samples thermally annealed at (**b**) 50 °C; (**c**) 200 °C; (**d**) 250 °C; and (**e**) 350 °C.

**Figure 2 nanomaterials-13-01775-f002:**
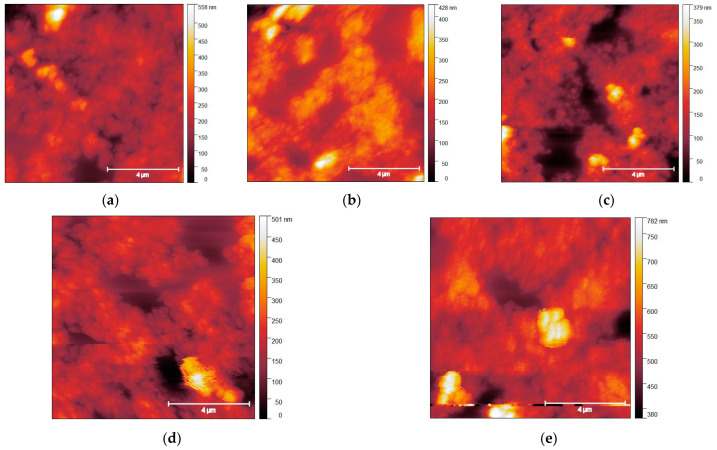
Typical AFM topography images of the (**a**) initial CZTS NC film and FLA-treated samples with energy densities of (**b**) 2 J/cm^2^; (**c**) 6 J/cm^2^; (**d**) 8 J/cm^2^; (**e**) 12 J/cm^2^.

**Figure 3 nanomaterials-13-01775-f003:**
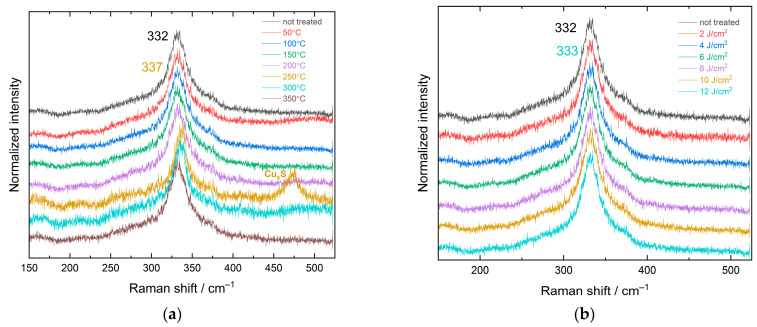
Raman spectra of CZTS NCs treated (**a**) with different annealing temperatures and (**b**) with different FLA energy densities.

**Figure 4 nanomaterials-13-01775-f004:**
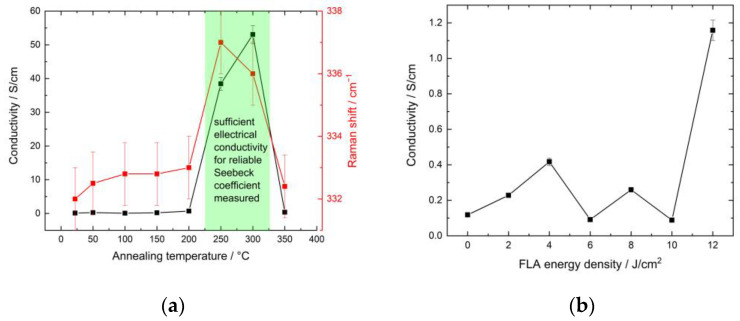
(**a**) Electrical conductivity and main Raman peak position of CZTS phase of CZTS NC films thermally annealed at different temperatures. (**b**) The dependence of electrical conductivity on the energy density of FLA treatment of the CZTS NC films.

## Data Availability

The data presented in this study are available on request from the corresponding author. The data are not publicly available due to some of the presented results being preliminary and still in the processing stage, and they will be the topic of a future paper.

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
