# Peer review of "Influence of Thermal and Flash-Lamp Annealing on the Thermoelectrical Properties of Cu2ZnSnS4 Nanocrystals Obtained by “Green” Colloidal Synthesis"

_nanomaterials, 2023, doi:10.3390/nano13111775_

Round 1

Reviewer 1 Report

In general, I appreciate the idea of combining photovoltaics and thermoelectrics. In practice, however, it is evident from this study that the idea did not work out at all, given the poor thermoelectric performance of these films.

For this to be published, the authors need to be more thorough and open in the discussion of the thermoelectric properties, both about their films and the literature. For example, papers like https://doi.org/10.1039/D1TA02978A need to be cited, and the data compared with.

Then, at minimum, a detailed outlook needs to be presented about how to improve the films to better match the literature, and make the project noteworthy.  Ideally the authors would already implement 1 or 2 of those ideas before a resubmission.

Author Response

Authors reply: we are grateful to the reviewer for careful reading of the manuscript and comments made for its improvement. We have taken the suggestions into account and added the following paragraph in the manuscript (page 8):

“In general, for all films of CZTS NCs obtained by the colloidal synthesis in this work the conductivity is much lower than the values commonly reported for bulk crystals [66] or polycrystalline films obtained by other methods [32]. The main reasons may be a poorer crystallinity of our films due to the mild conditions of both the NC synthesis and film fabrication. In addition, the films can contain some minor residuals of the ligands, not completely removed by centrifugation of the colloid and annealing of the film. The component composition of the CZTS NCs may also play an important role as well as the thickness and quality of the deposited films. Thus, to improve the conductivity of the CZTS NCs films obtained by such a synthesis method, the synthesis conditions should be further optimized (for example, further purification from ligands, different NC composition etc.), as well as the film deposition method (thicker and denser films are expected to show better conductivity). “

A systematic study of the above factors is not possible within the (short) time given for revision of the manuscript. Nevertheless, we performed several preliminary experiments in the meantime, in particular tried to improve the conductivity by making films with consecutive spin coating of multiple layers and variation of Cu content in the NCs, performed additional post-synthesis purification of the NC colloid, and deposited thick films by drop-casting (instead of thin spin-coated films). However, none of the implemented approaches resulted in noticeable improvements in conductivity, which was still not sufficient for Seebeck coefficient measurements without annealing of the film at high temperature. It is important to note here that for the successful synthesis of CZTS NCs with different copper contents it is not enough just to take a different volume of copper precursor, but other modifications of the synthesis conditions are also required, which could be a separate study. As we have established in the present work, the CZTS NC films can get electrically conductive upon annealing at certain temperatures (i.e. 250°C and 300°C), but their conductivity and Seebeck coefficient are found to be mainly determined by the CuxS phase. Therefore, improvement of the intrinsic conductivity and TE performance of CZTS in the films of NCs obtained by colloidal synthesis in aqueous solutions is an important task, which requires further in-depth investigations.

Going back to the point of reviewers comment, we would like to say that, in our opinion, based only on this study it is too early to judge whether thin films prepared from colloidal CZTS NCs synthesized at mild conditions are suitable for TE or hybrid TE-PV applications. In the scope of the present study we focused on establishing the relation between the electrical and TE characteristics of such films and their vibrational (Raman) spectra. Understanding such relation will facilitate more controllable tuning of the film structure for optimizing the electrical and TE characteristics in the following studies.

Reviewer 2 Report

The manuscript presents the heat treatment of colloidal prepared CZTS by conventional annealing method and FLA method, and shows the relations between the process parameters, microstructure, structural phase transition, and thermoelectric properties. From the investigation the authors gave the optimum conditions for possible applications of the material such as solar cells. The experiments were well described and the data analyses were also very detailed and informative, which makes the article deserve publication in the journal. I only have two questions which should be clarified before the publication: 1) as from the experimental part, you measured the thermoelectric properties up to 900K, very high temperature compared to 300°C, the microstructure and phase constitution in the film could be already different, how about the property stability? It is very important for applications. 2) Figure 4 shows the conductivity and Raman shift as a function of annealing temperature, does the conductivity depend on the temperature? If it does, the conductivity in Figure 4 is from which temperature? Or the maximum value? How about the Seebeck coefficient and/or the power factor changing on annealing temperature?

Author Response

Authors reply: we are grateful to the reviewer for careful reading of the manuscript and comments made for its improvement.

  • 900 K is the upper range of the device, but all of the TE measurements were performed at the 310K. This temperature was chosen for thermoelectrical characterization as minimal temperature that can be controlled by the device to exclude the influence of room temperature fluctuations. The higher temperatures can affect the CZTS NCs structure and phase, so to clearly differentiate the influence of temperature and FLA annealing on the thermoelectrical properties of CZTS NCs from the influence of measurement conditions, the temperature of 310 K was chosen. The sentence “The characterization was performed at 310 K to avoid any influence of the measurement conditions on the structure and clearly separate the influence of different annealing conditions.” is added to the manuscript.
  • The electrical conductivity was also measured at 310 K. Unfortunately, the conductivity of the samples annealed in the temperature range from 50 °C to 200 °C is too low for sufficient Seebeck coefficient measurements. That is why we cannot make any conclusions about changes, except of samples annealed at 250 °C and 300 °C, where the conductivity allows the Seebeck coefficient to be measured as described in the manuscript.

Reviewer 3 Report

The introduction must be improved. There are a lot of lumped references. The authors have to avoid them and brief description each paper against the topic of the paper. 

Author Response

We are grateful to the reviewer for careful reading of the manuscript and comments made for its improvement. 

We splitted the lumped references in the introduction and specified the particular results obtained in them:

“Recently CZTS has attracted the attention also in terms of thermoelectric properties  [21–31]. Both bulk crystals [26], polycrystalline layers [25,27] and nanocrystals [22] are considered. Controllable deviations from Cu2ZnSnS4 stoichiometry [23] and cation substitution [28] was suggested as one of the parameters to be tuned for the optimization of the CZTS TE performance [31]. "

Reviewer 4 Report

Comments to Author: 

The manuscript by Havryliuk et al. entitled: 

"Influence of thermal and flash lamp annealing on the thermoelectrical properties of Cu2ZnSnS4 nanocrystals obtained by "green" colloidal synthesis" 

reports the study of Cu2ZnSnS4 nanocrystaline thin films prepared by spin-coating from a colloid obtained by “green” synthesis. Different thermal or flash-lamp annealing procedures were made, with Raman spectroscopy and AFM being used in their chemical and surface characterization. The electrical (electrical resistivity) and thermoelectrical (Seebeck coefficient) properties of such films were also investigated. Albeit being an interesting and manuscript, it contains several lacks and mistakes and therefore it cannot be accepted for publication in “Nanomaterials” in its current form.

In particular:

Information that can be obtained from thermogravimetric analysis, SEM observations and powder XRD investigations of the materials, which would complement, confirm and reinforce the presented conclusions, is missing. We strongly suggest the Authors to make such kind of measurements.

Author Response

We are grateful to the reviewer for careful reading of the manuscript and comments made for its improvement. 

We agree that performing more characterizations can give some additional information about our samples. We already performed XRD measurements, but due to the small films thickness (in range of 100 nm) no informative diffraction pattern could be obtained. For all samples the signals from CZTS NCs overlap with those of the glass substrate. Even using grazing incidence geometry with 1° angle did not help. For the samples, where the CuxS phase is present, the features in the diffractograms at the positions corresponding to the CuxS are observed, but the signal-to-noise ratio is not sufficient for further analysis. This issue was one of the main reasons for employing Raman scattering as a main characterization technique. Owing to the possibility of using resonant excitation, Raman spectra of good quality can be obtained even from such thin films. The morphology of the films was explored by AFM in our work. Performing SEM is not expected to provide more information. Also from our point of view it is not expected that thermogravimetric analysis will provide any additional information useful in the scope of the paper.

Reviewer 5 Report

The reviewed manuscript is provided with a very interesting introduction on the use of heat generated when solar cells are irradiated for use in thermoelectric microgenerators. 

However, the substantive content of this work, concerning the change of thermoelectric properties of Cu2ZnSnS4 thin layers, is much weaker - many times smaller Seebeck coefficient and electrical conductivity in comparison with monocrystalline or polycrystalline films. Probably the method of fabrication of CZTS layers used by Authors will not be used on a larger scale in the future. Therefore, the conclusions of the Authors connected with annealing of these layers will not have much practical significance.

In the article there is no relevant information on the planar sizes of the fabricated structures. There is also no information on how the Authors obtained the flash annealing lamp with a specific energy density, whether the radiation of the used Xenon lamp was homogeneous, and if so, on what area. 

Based on above remarks I propose to reconsider publishing of this manuscript after major revision.

Author Response

We are grateful to the reviewer for careful reading of the manuscript and comments made for its improvement. 

1) Our motivation for this study was the following: on the one hand, we have expertise in colloidal semiconductor NCs, which are intensively produced and studied nowadays as promising and convenient source of material for producing thin-film absorbers and thermoelectric layers; on the other hand, we have applied an available commercial setup for large-scale fabrication of thin films by photonic annealing. Therefore, the investigation of combining these two technologies for producing functional thin films from colloidal NCs appears for us very well substantiated. We agree with the reviewer that the performance obtained so far is not satisfactory for application. However, the purpose of our first/preliminary experiments is to highlight the existing challenges and suggest solutions rather than to provide a recipe of preparing device-grade active layers for thermoelectric or photovoltaic application.

2) We used a commercial setup to perform flash-lamp annealing treatment in our work, as specified in the Materials and Methods section. the next sentences are added: “The values of the power density, which we used (2 J/cm2, 6 J/cm2 etc.) are known from its specifications. As the illuminated area (» 40x750 mm) is much larger than the size of our samples (» 4x25 mm), therefore the illumination is expected to be homogeneous for each sample”.

Round 2

Reviewer 4 Report

Very thin films (down to some units of nm) have been studyied using X-ray diffraction, either in the BB (e.g. https://doi.org/10.1116/1.2166860), the PB (e.g. http://dx.doi.org/10.1063/1.4903165) or in the GI (e.g. doi:10.1107/S0021889812000908, https://doi.org/10.1063/1.3251205) geometries, using conventional or synchrotron radiation. Therefore, ist is unlikely that the inexistance of diffraction peaks came from the technique, being most probably due to que quality of the samples. The measured data should be provided and must be discussed in the manuscript.

SEM observations can give additional information, in particular about the composition, if BSE detectors are used and if composition maps are collected. SE can give information about the surface morphology that can be compared with the AFM data (confirming it or not). We strongly advise the authors to make and discuss such kind of observations.

Author Response

We agree that in general good diffractograms can be obtained from thinner films than ours. However, there are a number of factors that play a very significant role. First, the substrate, which can influence the formation of crystallites of certain orientation, as well as the signal from the substrate should not overlap with the signal from the sample. Secondly, the method of producing the film plays a crucial role. Thus the film produced by epitaxy at high temperature in vacuum will have long-range crystallinity that ensures good XRD reflexes even at a few-nm film thickness, while the film consisting of few-nm NCs will not produce comparable XRD patterns even for tens of nm film thickness. Finally, the different materials of the film itself give a completely different response in XRD. The examples chosen by the Reviewer of good XRD spectra from the literature were obtained on completely different material, substrate, and deposition method than the films investigated in our work, and therefore cannot be compared. Besides, the Reviewer continues to insist on SEM measurements. However, from the authors point of view, the topography obtained with AFM does not need any further verification as it is a more direct method of topography measurement than SEM. In addition, due to the low conductivity of our films, SEM measurement may need deposition of a metal layer which could affect the film properties. Determination of the elemental composition using EDX, from the authors point of view, is useless in the context of this article, since it an average result over large number of NCs and can neither establish the reason of low conductivity nor show the localization of inclusions of secondary phases. Therefore, from our point of view, the Reviewer's comments are not relevant to the scope of our paper, but only show his/her own preference for SEM over other techniques.

Reviewer 5 Report

I did not notice an improvement in the results of the research conducted by the authors. The results are very far from the thermoelectric properties of single crystals. Also, the use of flash annealing lamps was practically unnecessarily described because the improvement in the conductivity of the layers was negligible and the authors did not mention changes in the Seebeck coefficient. I believe that the subject can be returned to if the authors note a significant improvement in the thermoelectric properties of the layers they have produced.

Author Response

Here, the Reviewer notes that our thermoelectric results are far from the results for single crystal. However, as we noted above and as indicated in the text of the paper, that it is not relevant to compare certain properties of the samples obtained by completely different ways, particularly the CZTS single crystal and thin film composed of nm-large CZTS NCs. Of course, more research should be performed to improve the conductivity of thin films of CZTS nanocrystals obtained by colloidal synthesis, but it is beyond the scope of this article and, as noted above, may be a separate and much more extensive study. Regarding the comment about no need to mention flash annealing, we strongly disagree with the Reviewer. First, similar studies have never been done before, secondly the information about weak influence of various annealing types on electrical conductivity of CZTS films even with improvement of their crystallinity, which follows from Raman, is already an interesting result and can be useful for other researchers working in the same field. And thirdly, it is also interesting that the secondary phase of copper sulphide, which is responsible for the measurable thermoelectric properties of those samples, does not appear at flash annealing and only at certain temperatures of the thermal annealing. Furthermore, it is too early to speak about uselessness of flash annealing until electrical conductivity is improved to the level allowing to measure thermoelectric properties of CZTS nanocrystals obtained by colloidal synthesis.